# What Promotes Natural Forest Protection and Restoration? Insights from the Perspective of Multiple Parties

**Jinkai Ke †, Ruolin Sun †, Yifei Ma and Caihong Zhang ***

School of Economics and Management, Beijing Forestry University, Beijing 100083, China
* Correspondence: zhangcaihong650215@bjfu.edu.cn; Tel.: +86-1370-1300-549
† These authors contributed equally to this work.

**Abstract:** The natural forest protection and restoration (NFPR) system is imperfect due to contradictions between the objectives of natural forest protection and the reality of situations, outdated cultivation concepts, conflicting interests among participating parties, and the lack of regulation guarantees and assessment criteria. These problems are not only common in China but also in international forest protection. As the NFPR system is more focused on the protection of natural forests, the level of natural forest restoration in China has been poor, with low natural forest quality and forest productivity. At the same time, the value of natural forest ecosystem services does not match the demand of farmers, forest management, and other multiple participating parties. As a result, except for the government, other multiple parties lack the intrinsic motivation to participate in NFPR, ultimately forming a sustainable management dilemma. Under the institutional analysis and development (IAD) framework, the objective of this research was to explore the influencing factors and outcomes of the participation of multiple parties in NFPR and to construct a multiple parties' participation mechanism for solving this dilemma. This research found that among external variables, multiple parties' characteristics, biophysical conditions, attributes of community, and rules-in-use jointly influence and constitute the driving mechanism of multiple parties' participation in NFPR. The rules-in-use directly impact the participation action scenario and regulate the other three external variables. Various factors and mechanisms in NFPR interact in the action space and produce outcomes that create positive incentives for each external variable, thus promoting the whole mechanism to achieve a virtuous cycle of sustainable management. This study provides a theoretical contribution to understanding the behavior of multiple parties participating in NFPR.

**Keywords:** natural forest protection and restoration (NFPR); multiple parties; institutional analysis and development (IAD) framework





## 1. Introduction

As the primary component and essence of forest resources, natural forests are a terrestrial ecosystem with the most enduring communities and the greatest diversity of life on earth [1]. The 2021 China Ecological Environment Status Bulletin states that China's forest cover was 23.04% in 2021. The forest stock was 17.56 billion cubic meters, including 14.108 billion cubic meters of natural forest stock and 3.452 billion cubic meters of planted forest stock [2]. China's natural forest stock accounts for 80.34% of the entire forest, and the protection and restoration of natural forests have a great impact on the entire forest ecosystem. The important role of forests in improving the climate is clearly defined in the Paris Agreement, which encourages developing countries to restore and sustainably manage their forests [3]. The thorough protection and restoration of natural forests are of enormous importance for the development of ecological civilization and the improvement of the global environment [4]. Natural forests are forests that originate from a natural state rather than from artificial cultivation, that remain relatively natural without disturbance or, to a lesser extent, recover naturally after disturbance, including residual primary forests

or over-harvested forests in former natural forest areas, natural secondary forests and degraded forests of varying degrees, and sparse woods. Before 1998, natural forests played a leading role in the production of forest products in China and were an important material basis for achieving sustainable social and economic development [5]. However, on the one hand, natural forest resources were insufficient in total and of poor quality; on the other hand, during this period, China's natural forest resources had long been in a state of over-harvesting and unreasonable management. Natural forest resources had been damaged, and used inefficiently, with backward protection and management [6], which caused a sharp decline in natural forest resources and the degradation of ecological functions, resulting in serious ecological and economic problems. It was not until 1998, following the devastating floods in China, that the country began to implement the Natural Forest Protection Project (NFPP) in the hope of accelerating the speed, improving the quality of natural forest ecological restoration, and ensuring regional ecological security and sustainable economic and social development [7]. Nevertheless, China's natural forests are scarce in quantity, weak in quality, and fragile in the ecosystem, with problems related to an imperfect protection system, low level of management, and insufficient participation from multiple parties, forming the basis for the sustainable management dilemma [8]. Effective participation from multiple parties is crucial to solving this dilemma; therefore, this study attempts to identify the influencing factors and effects of the participation of multiple parties in NFPR and to construct a mechanism for the participation of multiple parties in NFPR.

The ecosystem service function of natural forests as a public good has obvious externalities, and it was mainly supported by the government-led public financial act [9]. The United States, Canada, Japan, and some developed countries construct national parks and nature reserves, and Germany has adopted near-natural management while Australia and New Zealand carry out classified management [10]. China's emphasis on protection rather than restoration has led to a low willingness of multiple parties to participate in natural forest protection and restoration, which causes a sustainable management dilemma. Specifically, in the actual process of NFPR, the government apparatus engages through rules-in-use, while farmers, forest management (e.g., village economic collectives, state-owned forest farms), and other participating parties (e.g., NGOs, individuals) participate in NFPR through interactions such as forest certifications, forest carbon sink trading, and natural commercial forest redemption, thus enhancing natural forest quality. The ecological value brought by the enhanced natural forest quality provides an incentive for multiple parties to continue to participate in NFPR and for the system to continue to improve itself; subsequently, sustainable management of natural forests would thus be realized. Figure 1 depicts the above-mentioned process.

However, in practice, the government apparatus tends to just protect, and place strict restrictions on nurturing and harvesting that lead to a low degree and poor quality of forest cultivation and restoration. Meanwhile, the actual restoration process is not carried out in accordance with the technical guidelines and management principles, resulting in inefficient natural forest restoration and impeding the advancement of sustainable management [11]. Conversely, the supply value of natural forest ecosystem services does not correspond with the demand of multiple participating parties, and the market system is inadequate, which makes it difficult for social capital of a profit-making nature to obtain sustainable incentives, diminishing participant involvement and sustainable management [12]. In the end, the fact that farmers and forest management, among others, are reluctant to participate in NFPR through existing interaction patterns makes it hard for the government apparatus to achieve the sustainable management of natural forests.

At present, there are three main ways to effectively encourage the involvement of multiple parties in NFPR and to solve the sustainable management dilemma of natural forest resources. The first is to raise funds by issuing forest certifications [13] and carbon sink trading [14]; the second is to encourage various parties, such as citizens, legal persons, and other organizations, to donate, adopt, and volunteer [15]; and the third is to build

redemption systems for natural commercial forests in critical ecological zones [16]. Although these techniques have helped to foster participation in NFPR, they still have limited effects due to institutional flaws and market failures. In terms of institutions, the absence of effective oversight has given the government a tendency to protect to gain direct and instant results, rather than to restore. In terms of markets, the demand of the profit-seeking market capital is out of line with the potential value of natural forest ecosystem services, which is reluctant to engage in NFPR for the public good [17]. Therefore, based on institutional development and market soundness, a set of rules and standards, a reciprocity model or supervision, and a restraint mechanism are needed to improve the mutual trust among multiple participants and increase their willingness to resolve the sustainable management dilemma.

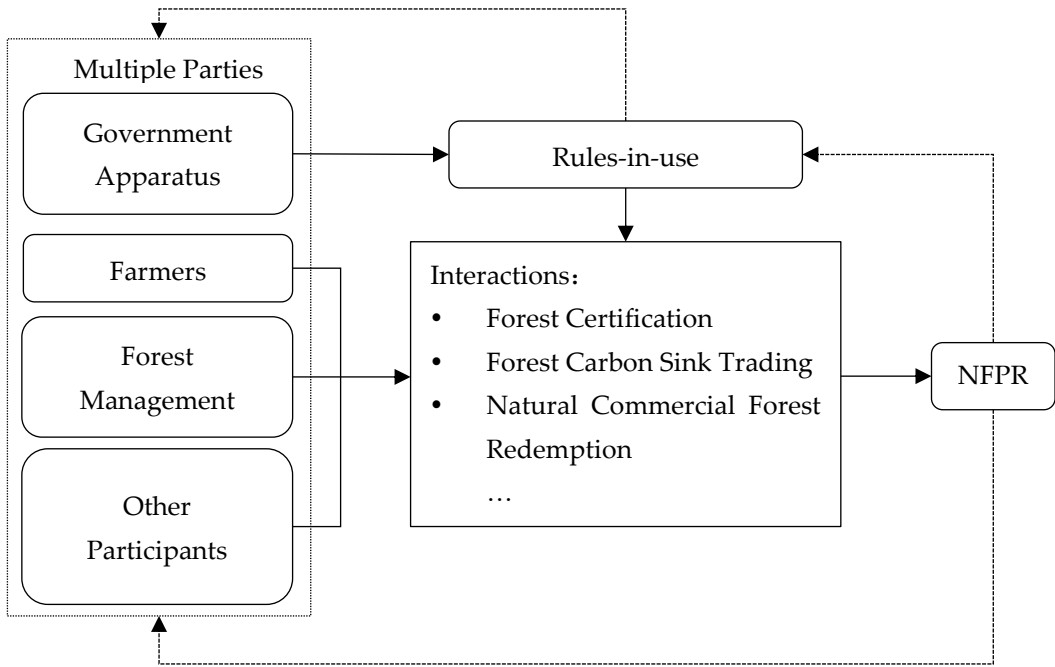

**Figure 1.** The process of the participation pf multiple parties in NFPR.

Therefore, this study is crucial to clarify the driving mechanisms of multiple participants and explain the logic behind their participation in NFPR. This study also contributes to the collective action theory. The study focuses on the mechanism of multiple participants in NFPR and uses the institutional analysis and development (IAD) framework to systematically explore the complex public governance mechanisms, expands the connotation of multiple participants in NFPR, and contributes a Chinese case to the collective action theory. As for practical contributions, this research provides new ideas for the development of NFPR systems in China and other countries.

## 2. Framework and Methodology

### 2.1. Framework

The IAD framework is an essential framework and classical theory for studying public policy [18]. According to Elinor Ostrom (2011), the IAD framework can identify factors and relationships in the actions of multiple parties, provides a structured approach to study the action logic of participating parties in different scenarios, and plays a vital role in solving the collective action dilemma of multiple parties [19].

In the original IAD framework, Ostrom (2009) classified external variables into three categories: biophysical conditions, attributes of the community, and rules-in-use [20]. Natural forest resources are closely related to biophysical conditions. At the same time, the participation of multiple parties in NFPR are motivated and constrained by rules-in-use, so

it is necessary to take biophysical conditions and rules-in-use as important external variables for analysis. In addition, attributes of the community mainly influence the behavior of multiple participants in terms of economic development and social development level. More importantly, attributes of the community can be utilized to examine homogeneous or heterogeneous characteristics among multiple parties as well as the influence on participation behavior. Therefore, this study extracts the characteristics of multiple parties based on attributes of the community as a separate variable and focuses on exploring the influence of homogeneous or heterogeneous characteristics on multiple participants' behavior. Under the IAD framework shown in Figure 2, this study further analyzes the action logic and driving mechanism of multiple participants in NFPR.

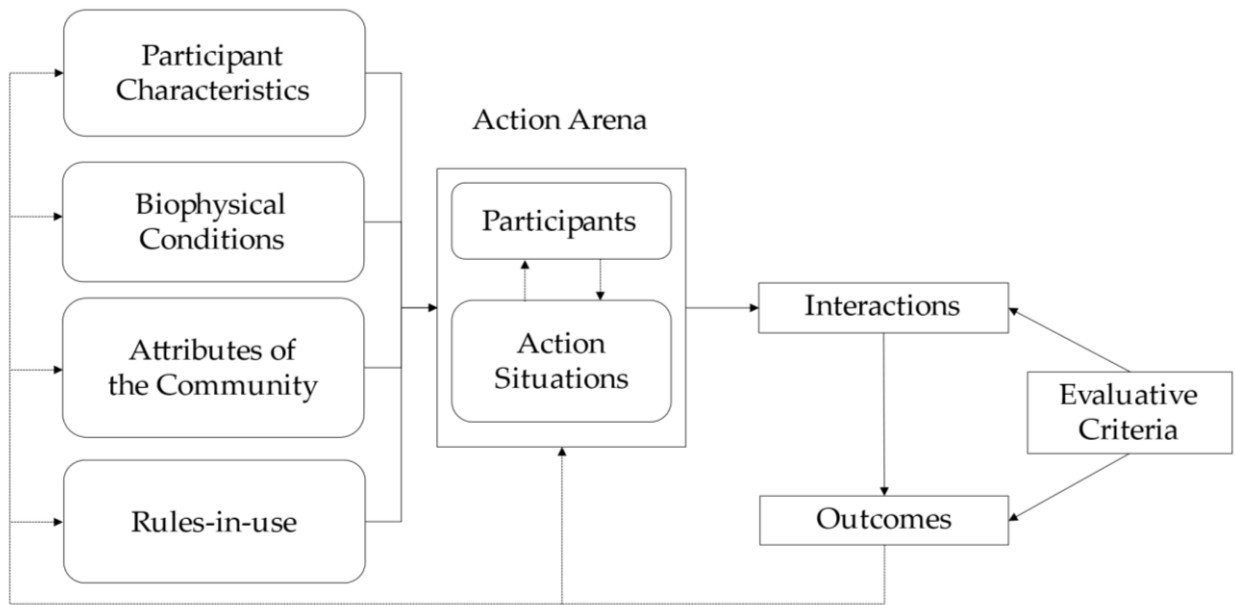

**Figure 2.** Institutional analysis and development framework.

### 2.2. Methodology

The primary method adopted was a literature study. We used Web of Science, CNKI, Elsevier, and other citation index databases. We searched keywords such as natural forest protection and restoration (NFPR), Natural Forest Protection Program (NFPP), and multiple parties. Considering the timeliness of the study, we selected literature from SSCI, SCI, CSSCI, and CSCD, since the second phase of the NFPP (since 2010) and obtained 86 articles for analysis. In general, the selected literature had the following characteristics: in terms of space, the research on NFPR was mainly concentrated in China; in terms of time, the early studies mainly focused on theories and specific practices of protection and restoration, and in recent years, more and more studies have been conducted on specific impact mechanisms; in terms of methodology, most studies adopted a quantitative approach with a few employing a case study or theoretical study.

## 3. Influencing Factors on Multiple Participants and NFPR

From the existing research, many influencing factors affect the participation of multiple parties in NFPR. From the macro perspective, there are factors like fairness, efficiency, participation, and sustainability [21]. From the micro perspective, there are factors including age [22], education level [23], household size [24], household income [25], biophysical conditions [26], and willingness to participate [27]. Under the IAD framework, this study classifies the influencing factors into four categories: multiple participant characteristics, biophysical conditions, attributes of the community, and rules-in-use. The specific factors included in each category are systematically sorted out through literature research.

*3.1. Participant Characteristics*

3.1.1. Characteristics of Farmers

Farmers are one of the main parties concerning NFPR, who participate in NFPR mainly through changes in the quantity, quality, and structural configuration of labor supply [28]. Farmers' participation behavior is guaranteed by reasonable and competent forest management, and the pertinent system serves as the foundation. The reasonable and effective participation behavior carried out by farmers is supposed to be in accordance with the corresponding system incentives [29]. In addition, other characteristics, such as farmers' education level, basic qualities, and future expectations, also significantly affect participation behavior [30]. Specifically, this is reflected in the effects of population, income, willingness, and the degree of part-time work.

(1) Farmer population. Studies have concluded that there is a negative influence between the farmer population and the quantity and quality of natural forests, thus reducing the quality of NFPR. Considering the cost of livelihoods, farmers tend to cut down forests to expand agricultural production and increase household income [31–33]. With NFPR projects and the implementation of policies such as comprehensive commercial logging bans, the impact of population growth on deforestation is gradually decreasing [34]. This process is mainly accompanied by a general increase in education level and quality of the labor force, and a decrease in the dependence of population growth on forest resources along with an increase in non-farm employment [35]. It has also been shown that population growth in China is no longer hazardous to resources and that a moderate increase in population will improve the quality of natural forests in conjunction with, for example, the development of technology [36]. In conclusion, when the cost of livelihood is high, the increase in the farmer population will inhibit the participation of farmers in NFPR; while if the level of education and technological development is high, the increase in the farmer population will have a positive effect on participation.

(2) Farmer income. After satisfying the basic material needs, farmers usually generate non-material life needs. The ecological benefits brought by natural forest ecosystems, such as environmental improvement and soil and water conservation, will become their incentives to participate in NFPR [30]. Therefore, increasing farmers' income will promote their participation in NFPR.

(3) Farmer's willingness. Farmers' willingness to participate in NFPR relates to intergenerational transmission. Therefore, this requires long-term sustainable influence through the formulation of local regulations and the implementation of public welfare activities. The opportunity cost of farmers' behavior will have an important impact on their willingness to participate. Their willingness to participate rises when they are compensated with a relatively large portion of their annual household income per capita. Therefore, the stronger the willingness of farmers to participate in ecological protection, the more likely they are to participate in NFPR.

(4) Farmer's part-time work. Part-time work is a rational choice for farmers to maximize income, which can be changes in family income structure and the participation behavior in NFPR. At the same time, it will also affect the change in the labor force employment structure and further changes in the industrial structure, which will directly influence the farmer's participation behavior. Additionally, farmers' part-time work will indirectly affect their participation in NFPR through changes in opportunity costs, such as labor time and transportation [37].

3.1.2. Characteristics of Forest Management

Forest management are also one of the main participants in NFPR. According to the division of forest property rights ownership, forest management in China consist of village collective organizations, forest farmers, and state-owned forest organizations that operate state-owned forests. Forest management organizations' degree of trust among their members, regulatory mechanism, and method of participation all impact participation in

NFPR [38]. In addition, the leader's characteristics, such as social status can also influence forest governance and thus the degree of participation [39].

### 3.1.3. Characteristics of Government Apparatus

The participation of the government apparatus is the most important mechanism of NFPR [40], as the central government regulates NFPR through macro-policies and local governments to implement specific policies. In the existing natural NFPR system, the government provides incentives for other parties, which reduces information asymmetry and externalities, and designs corresponding mechanisms for the interests of basic-level participants, such as farmers and forest industry workers [41].

### 3.1.4. Characteristics of Other Participants

In addition, some social organizations and individuals also participate in NFPR, and they are highly sensitive to the participation patterns. The supply of public services, such as natural forest resources, has gradually evolved from a single government provision prioritizing efficiency to a cooperative participation of multiple parties. Currently, the other participants are roughly divided into social organizations, enterprises and individuals, and function through government, market and voluntary supply [42]. The participation of multiple parties will be encouraged by more participants, diversified ways of participation and stable institutional rules [43].

### *3.2. Biophysical Conditions*
### 3.2.1. Forest Land Conditions

Forest land is the main carrier of NFPR, and the quantity and quality of forest land are prerequisites for the participation of multiple parties in NFPR. The quantity of forest land can directly affect the scale of natural forest resources management, protection cost, and restoration efficiency, and the quality of forest land can directly influence the structure and optimization of natural forests and management systems. Due to the externality of forest resources, relatively poor woodland conditions are mainly involved in protection and restoration by government. In contrast, relatively good woodland conditions will attract other multiple parties [29]. In addition to quantity and quality, cultivated area and degree of forest fragmentation also play an indirect role. Among them, there is a substitution effect between cultivated and forest land areas, and the agricultural expansion process is often accompanied by forest land loss [34]. The higher the degree of forest fragmentation, the lower the integrity of forest ecosystems [44], and the ecological functions of forests will be significantly weakened, leading to a consequent decrease in the quality of natural forests.

### 3.2.2. Forest Conditions

Forest conditions refer to the structure and growth conditions of forest trees. Forest conditions directly affect the coordination and sustainability of the natural forest resource management system, affecting multiple parties' participation behavior [29]. Moreover, besides the forest conditions of natural forests, the coverage of economic forests also has an indirect effect. Yan Ruhe (2020) found that there is a competitive effect between natural forests and economic forests. At the same time, there is a substitution effect between the ecological effect of natural forests and the economic effect of economic forests. Ultimately, expanding economic forests will improve the quality of natural forests but will reduce the natural forest coverage [45].

### 3.2.3. Climate Conditions

The different climate is an objective factor affecting the quality of natural forests. For example, two climatic factors, temperature and precipitation, can impact soil wind erosion, habitat quality, and forest carbon storage [46]. Specifically, the amount of precipitation and the effective temperature accumulation can directly affect the type of natural forest

vegetation, the type of natural forest ecosystems, and the way natural forests are protected and restored [29].

### 3.2.4. Geographic Location

Since different regions have significant differences in geographic situation, climate, geomorphology and institutions, the influencing factors for participation in NFPR vary from region to region. For instance, regarding the geographical situation, Chen (2007) argued that the northwestern region is arid and less rainy than the southeastern region of China, with severe desertification and salinization. Most hills and deserts in the northwestern region generally have low forest quality and relatively low NFPR participation [35]. From the watershed perspective, Sun, Zhen and Wang (2017) found that the degree of ecosystem service enhancement in the Yellow River Basin was significantly higher than that in the Yangtze River Basin after the implementation of the NFPP [47].

### 3.2.5. Natural Disasters in Forest

Natural disasters in forests can lead to a decrease in forest area, storage volume and a decrease in forest density, lowering natural forest resource quality, which indirectly reduces the willingness of multiple parties to participate [29]. Some studies concerning forest land resource renewal show that forest fires can promote forest land renewal and thus increase natural forest resources [28].

### *3.3. Attributes of the Community*
### 3.3.1. Economic Development Level

The economic development level influences the willingness to participate and specific patterns. Usually, there is a displacement relationship between environmental resources and economic growth, especially in rural areas, where the participation behavior of multiple parties is often influenced by livelihood needs and environmental pressures [36]. However, an increase in economic development can indirectly boost the participation of multiple parties through technological progress and human capital accumulation [48]. Additionally, when economic development levels rise, investments in natural forests will be attracted, which will help to encourage participation from multiple parties.

### 3.3.2. Industrial Restructuring

Industrial restructuring essentially changes the rough forest industry structure and affects multiple parties in various aspects, such as labor force scale and industrial output value. Industrial restructuring directly impacts the forestry industry's percentage, and when that percentage drops, the forest cover, the ecological forest area, and the quality of the natural forest all improve [49]. Meanwhile, industrial restructuring would transform forest age, origin, and species structure to achieve forest transformation [50]. The industrial restructuring that tends to be reasonable also promotes the improvement of natural forest quality while motivating multiple parties to participate in NFPR.

### 3.3.3. Forest Carbon Sink Market

The development of the forest carbon sink market will increase forest stocks, expand carbon storage, and promote the participation of multiple parties [51]. Among them, the size of exploitable natural forest land directly determines the amount of carbon sink. Forest land with larger offsets and stable average transaction prices is more attractive to multiple parties [52]. Through the carbon sink potential evaluation method, Chen et al. (2021) concluded that there is a greater potential for developing natural secondary forests to operate as carbon sink projects [14], which can achieve the purpose of multiple participants in NFPR.

### 3.3.4. Agricultural Production Efficiency

The improvement in agricultural production efficiency will significantly improve the quality of NFPR. The improvement in agricultural productivity is mainly influenced by improving education and thus human capital development; guiding social capital and financial capital into natural forest-intensive areas, such as villages; providing financial products, such as agricultural credit, developing village transportation, and improving local infrastructure, thus indirectly improving the quality of natural forests and influencing the participation of multiple parties [53–55].

### 3.3.5. Customs and Culture

In some remote ethnic minority gathering areas, where forest resources have long been the basis of their livelihoods, people have formed traditional NFPR methods. There are also some areas where ethnic minorities revere nature and forests, and so these local customs and culture have also become one of the influencing factors for the participation of multiple parties in NFPR [35].

### *3.4. Rules-in-Use*

Shen's (2009) research discussed the connection between natural forest protection and sustainable forest management, emphasizing the balance between the economic, social and ecological benefits, the integration of natural forest management objectives, and the development of a sustainable system to protect natural forests [11]. One of the most important influencing factors to encourage the involvement of multiple parties in protection and restoration is the rules-in-use.

### 3.4.1. Direct Rules

(1) Direct investment. Direct investment improves the quality of natural forests through afforestation, nurturing and management, optimizing labor supply and improving scientific and technological innovation, thus promoting the development of a non-wood economy. The substitution effect of other products on natural forest resources indirectly improves the quality of natural forests [48]. Additionally, a more equitable and efficient direct investment policy would promote the participation of multiple parties [56].

(2) Logging ban policy. Among the studies of NFPR policies, the logging ban policy has been a hot topic and the focus of much research, and its effects are diverse. From an industrial point of view, the complete cessation of commercial logging in natural forests, as a Pareto improvement, does not have a large impact on timber production [51]; from an ecological point of view, policies such as logging bans and closing hillsides to facilitate afforestation combined with techniques such as artificial regeneration and forest tending can significantly improve the quality of natural forests [57]. The social welfare enhancement promoted by the logging ban will boost the participation of multiple parties [58]. Some studies also argue that although the logging ban policy improves ecological benefits, the reduced economic benefits will inhibit the motivation of participants [28]. For example, Ke, Zhu and Yuan (2018), from a long-term perspective, argued that the logging ban policy should combine with the discretionary policy to avoid the ineffective use of natural forest resources [59].

### 3.4.2. Indirect Rules

Indirect rules are mainly made by the government to provides tax incentives for forestry, indirect investments, including forest management fees, policy-based sociological expenditure subsidies, etc. [28], and effective institutional protection systems such as assessment guidelines, laws and regulations [60]. These indirect rules generate economic support, social subsidies and ecological protection to natural forest areas and form a counterpart mechanism by government to promote the participation behavior of other parties.

### 3.4.3. Organizational Rules

(1) Ecosystem management. To manage the natural forest ecosystem, it is required to scientifically update human-made management practices to promote stand renewal and expansion [61], and apply forest management practices to increase the availability of ecological service goods [42].

(2) Contract management. The contract responsibility system of natural forest resources can effectively improve the quality of NFPR, and provide an opportunity for multiple participants. The contracted participants are responsible for managing, protecting, cultivating and operating natural forest resources in the contracted areas through forest management. They have clear responsibilities for the indiscriminate cutting and destruction of vegetation. Contracting increases the income of participants, optimizes the allocation of natural forest resources and realizes the sustainable management and development of natural forest resources [62].

(3) Division of management zones. Due to the unreasonable division standards of management zones, the difference in forest carbon density among management zones, such as key public welfare forests, general public welfare forests, and commercial forests, is insignificant, indicating that the overall protection and restoration efficiency is low. A reasonable division of management zones will increase the willingness of multiple parties to participate [63].

(4) Collective forest tenure reform. Collective forest tenure reform can enhance forest management, expand forest areas and accumulation, and promote the participation of multiple parties to a certain extent [64]. However, Zhang Han (2022) analyzed data from the institutional role channel that such reform's impact on forest quality development is not immediately apparent [65].

### 3.4.4. Incentive Rules

Fortmann et al. (1991) argued that exclusive management is the most effective way to protect forest resources, and sustainable management is created through appropriate incentive mechanisms [66]. The fundamental concept of substituting national economic capital with ecological capital has emerged in China due to the ongoing development of ecological civilization. Numerous ecological compensation projects have built a compatible mechanism of government and market, central and local incentives [67]. Ecological compensation is an incentive mechanism for NFPR, which can effectively motivate multiple participants by increasing income [68]. In recent years, more studies have been conducted on mechanism design and multiple payment methods [69].

## 4. Effects of Multiple Participants in NFPR

The third part of this paper analyses the key influencing factors of the participation of multiple parties in NFPR. The fourth part investigates the influencing effects of the participants in NFPR to elucidate the logical relationship between causes and consequences. Based on this, the fifth part uses the IAD framework to collate and summarize the logical connection between causes and consequences. Overall, there are two types of effects from participation in NFPR: direct effects and indirect effects. Direct effects include ecological benefits, the restoration and improvement of natural forest ecosystem services, and increased carbon storage and sink. In contrast, indirect effects include raising farmers' income, promoting farmer employment, facilitating forest enterprise development, and improving environmental health.

### *4.1. Direct Effects*

#### 4.1.1. Ecological Benefit

Protecting and restoring natural forests can help increase forest canopy cover and thereby better protect water quality, regulate water quantity and protect and enhance biodiversity [70]. Tree species and geographical situations further decide specific benefits: broad-leaved species yield greater value than coniferous species [71]; the total value of

ecological benefits of the NFPP in the Yellow River Basin increases by 95.5 billion USD per year, which is equivalent to 3.53 times the entire project investment, significantly higher than other regions [72].

### 4.1.2. Restoration and Improvement of Natural Forest Ecosystem Services

Six aspects of NFPR are primarily responsible for the restoration and improvement of natural forest ecosystem services: water conservation, carbon fixation and oxygen release, soil conservation, nutrient accumulation, purification of the atmospheric environment, and biodiversity protection [73]. Following the natural forest ecosystem's succession law, the stability of natural forest ecosystems and their ecological services can be improved by cultivating existing woodlands, adjusting the stand structure, creating compound and heterogeneous forests, and enhancing the biodiversity of forest areas [46]. The relevant institutional measures have led to a continuous increase in forest area and a continuous improvement in the value of natural forest ecosystem services [74].

### 4.1.3. Increase in Carbon Stock and Carbon Sinks

Natural forests are now China's primary source of forest carbon sinks due to an increasing trend in the carbon stock in NFPP regions [75]. The variation in carbon stocks in natural forests depend on the age group [76]. In particular, given the same input conditions, middle-aged forest carbon sinks grow more than young forest carbon sinks, and restoration measures, to some extent, have a higher carbon sink augmentation than forest planting measures [77]. Additionally, forest conversion, a low-cost and efficient way to improve carbon sinks, can provide a greater quantity of carbon sinks with comparatively less forest area [62].

### *4.2. Indirect Effects*

The indirect effects of NFPR are mainly reflected in the economic and social aspects, specifically in farmers' income, farmer employment, forest enterprise development, and environmental health.

### 4.2.1. Farmers' Income

Farm households now have better living conditions, social security, and psychological health because of NFPR [78]. On the other hand, low afforestation subsidies have widened the gap between urban and rural areas by causing a decline in the economic standing of farm households, a low level of participation, and a considerable reduction in operating income [79].

### 4.2.2. Farmer Employment

NFPR has several initiatives to enhance the livelihoods of people in the project area, including a forest resources system of contracted responsibility, settlement for laid-off workers, and subsidies. Due to the decline in their long-term income, the workforce has transferred to non-forest industries, while the subsidy program has failed to increase their motivation to continue to work [80].

### 4.2.3. Forest Enterprise Development

Additional financial support affects an enterprise's productivity due to variations in its mechanism. The total factor productivity of state-owned forestry firms is specifically decreased by eco-efficiency compensation, whereas political and social subsidies help increase enterprises' total factor productivity [81].

### 4.2.4. Environmental Health

Environmental health issues can be resolved with the help of forest resources, and NFPR can enhance public health and lower disease rates [82]. First, by conserving soil and water, NFPR can lower flooding, boost groundwater supply, and improve air quality.

Second, the risk of zoonotic infections also decreases by reducing human contact with wildlife. Third, regulating the use of resources allows access to safe and healthy forest products [83].

All the influencing factors on Multiple Parties and NFPR and the effects of Multiple Parties in NFPR have been summarized in Appendix A.

## 5. Results

### 5.1. Influencing Factors and Outcomes of Multiple Participants in NFPR

The influencing factors and outputs of the participation of multiple parties in NFPR are shown in Figure 3. The participating behavior of multiple parties in NFPR is mainly related to four groups of external variables: multiple participant characteristics, biophysical conditions, attributes of the community, and rules-in-use, and there are various specific variables in each external variable. Multiple participant characteristics include characteristics of farmers, forest management, government apparatus, and other participating parties; biophysical conditions include forest land conditions, forest conditions, climate conditions, geographic location, and natural disasters in the forest; attributes of the community include economic development level, industrial restructuring, agricultural production efficiency, forest carbon sink market, and customs and culture; rules-in-use include direct rules, indirect rules, organizational rules, and incentive rules. These influencing factors constitute the action space for multiple parties to participate in NFPR in different action scenarios. In the different scenarios, the multiple parties, under the supervision of evaluation criteria, make rational choices of forest certifications, carbon sink trading, natural commodity forest redemption, etc., to directly influence ecological benefits, ecosystem service restoration improvement, carbon storage, and carbon sink increase, as well as to indirectly influence farmers' income, farmer employment, forest enterprise development, and environmental health.

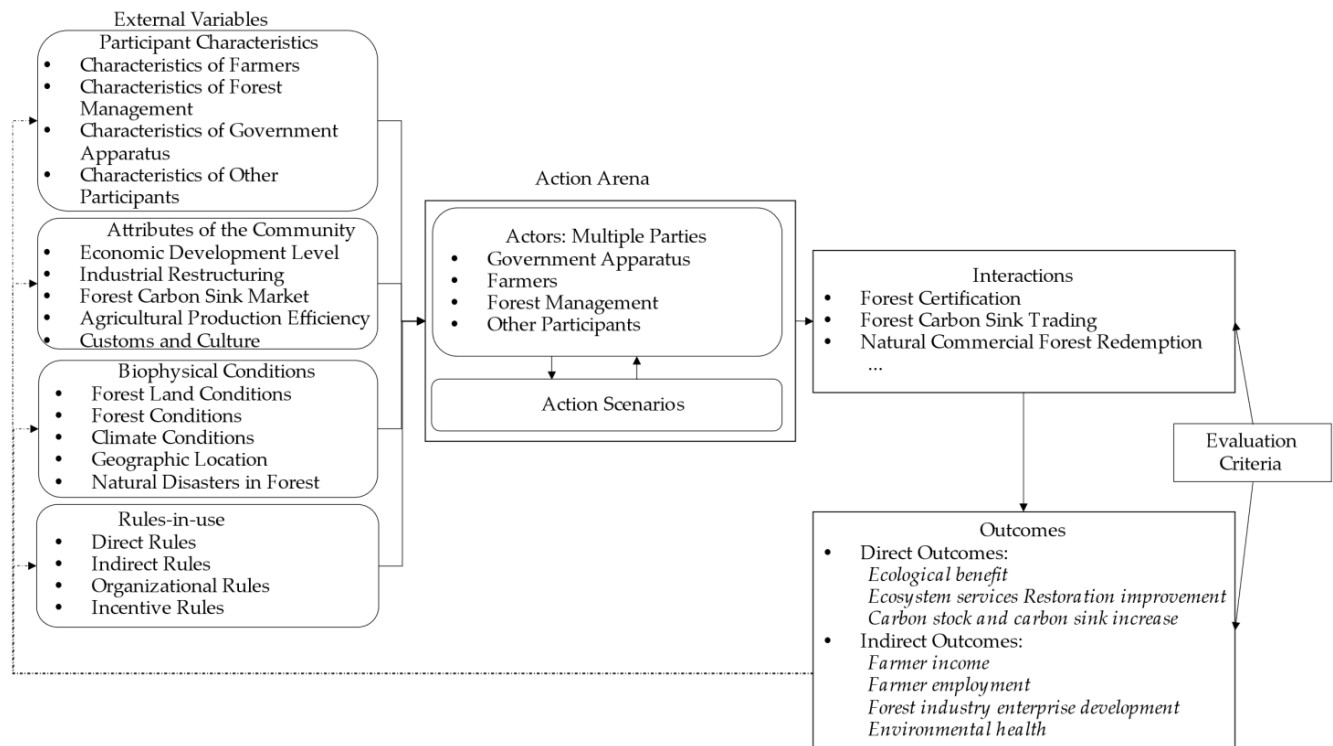

**Figure 3.** Influencing factors and outcomes of multiple parties' participation in NFPR.

### 5.2. The Effect of External Variables

From the perspective of sustainable management, multiple parties are sensitive to the lack of benefits in participating in the NFPR system due to regulatory restrictions,

ambiguous outputs, and insufficient oversight, causing a sustainable management dilemma. Consequently, the actual participation process is determined by complex influencing factors and mechanisms. Through the literature analysis, we can see that the participation of multiple parties in NFPR is affected by multiple influencing factors, which ultimately constitute the action scenario of multiple participants. The analysis of multiple influencing factors reveals that, in terms of the characteristics of multiple parties, farmers' income and willingness play a positive role, the influence of farmer population may be positive or negative, and the influence of farmers' part-time work is negative. The influence of the characteristics of the forest management is yet uncertain, and the influence of the government apparatus and other multiple participants is positive. Regarding biophysical conditions, the effects of forest land conditions, forest conditions, and climate conditions are positive, the effects of geographic location are uncertain, and natural disasters in the forest have negative effects but may also be uncertain. As for attributes of the community, the effects of the forest carbon sink market, agricultural production efficiency, and local customs and culture are positive, and the effects of economic development level and industrial restructuring may be positive or negative. In terms of rules-in-use, the influence of direct investment is positive in the direct rules, the influence of logging ban policy may be positive or negative; the influence of indirect rules and incentive rules are positive; the effects of ecosystem management, contract management, and collective forest tenure reform are positive in the organizational rules, and the effect of the division of management zones is uncertain.

The variable rules-in-use has a significant impact on the participation of multiple parties in NFPR [84]. The engagement of rules-in-use on the participation behavior of multiple parties is formed mainly through structural influences on the action space and other external variables. These structural influences constitute the underlying incentives and constraints for the participation of multiple parties in NFPR. Rules-in-use is more versatile and varied than the other three external variables. In other words, any element in the rule-in-use variable has the potential to affect at least one action scenario in the action space, and its flexibility can control the impact of other external variables on the action space [85].

*5.3. The Mechanism of Multiple Participants in NFPR*

In the course of the NFPR, both direct and indirect rules can encourage the renewal of forest conditions and enhance forest quality. Organizational rules, such as ecosystem management, contract management, and collective forest tenure reform can promote the participation of forest management and government apparatus, while improving climate conditions, raising the economic development level, and optimizing industrial restructuring. Incentive rules can guarantee farmers' income and increase their willingness to participate. Therefore, rules-in-use can significantly regulate the behavior of other external variables on the participation of multiple parties in NFPR, and can play a certain role in promoting the participation of multiple parties in NFPR through interaction patterns, such as forest certifications, carbon sink trading, and natural commodity forest redemption, so as to achieve the restoration and improvement of ecosystem services, increase carbon storage and carbon sinks, and increase ecological and economic social benefits. The whole process is monitored through the construction of evaluation criteria for NFPR. Finally, the positive incentives of these outputs are fed back to the external variables under the supervision of the evaluation criteria. Eventually, the whole system achieves a virtuous cycle of sustainable management of NFPR. The mechanism for the participation of multiple parties in NFPR is shown in Figure 4.

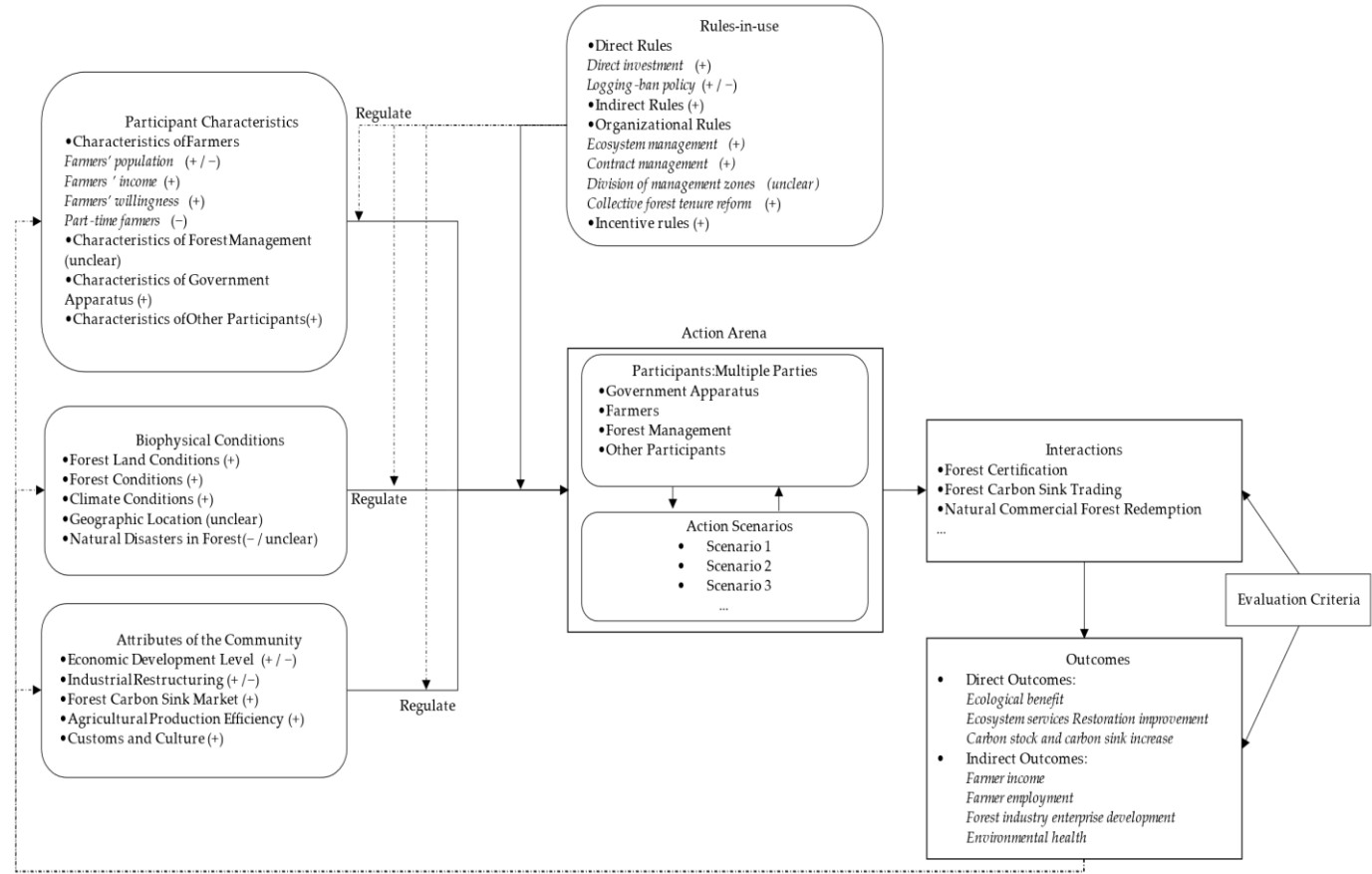

**Figure 4.** The mechanism for the participation of multiple parties in NFPR.

## 6. Discussion

This research attempts to use the IAD framework to analyze the literature relating to the participation of multiple parties in NFPR, explore the factors affecting the participation of multiple parties in NFPR, and the effects of the outputs of the NFPR system. The participation behavior of multiple parties results from the joint action of various external variables, and it is difficult for a single variable or factor to achieve the sustainable development of multiple parties' participation in NFPR. These factors also produce different impacts because all external variables jointly determine the various action scenarios for the participation of multiple parties in NFPR, and it is difficult for an individual factor to form the specific action scenario. Among all the variables, the rules-in-use variable is flexible and diverse, which can affect the characteristics of multiple parties, biophysical conditions, and attributes of the community from various perspectives and play an important regulatory role in the whole mechanism. The participation of multiple parties in NFPR can effectively solve several problems such as the sustainable management dilemma of natural forests, which manifests as the restoration and improvement of ecosystem services, the increase in carbon storage and carbon sinks, the rise in ecological, economic and social benefits, and the solution to the difficult problem, the sustainable scientific management of natural forests.

Because of the existing studies' limitations, some issues can be addressed in future studies. Firstly, to improve the research framework. This study focused on revealing the influence mechanism of multiple participants in NFPR from the attributes of four external variables. It only focused on the influence of rules-in-use on other variables, so the potential complex logical relationships between different influencing factors were not fully explored. In the future, more external variables can be supplemented and analyzed according to the scenarios, such as the effects of external variables on the whole mechanism and the mediating relationships between external variables. In addition, an advanced version of the

IAD framework, the social–ecological systems (SES) framework, can refine the specific roles between different influencing factors and clarify the mechanism of influence among all the variables [86]. Secondly, to conduct cutting-edge interdisciplinary research. The IAD framework provides a systematic framework for studying the behavior of multiple parties in NFPR. Researchers can conduct comprehensive research on cutting-edge interdisciplinary studies under the IAD framework by selecting natural resource factors, such as hydrology, soil, geology, and geomorphology, and then selecting economic and social factors, such as market and infrastructure construction. Third, to enrich research on the spatial–temporal differences and dynamic evolution. NFPR is a long process with its factors and mechanisms changing with time, and the specific mechanism varies from region to region. Therefore, future research needs to focus on cross-time and cross-regional research and analyze the mechanism's spatial variability and temporal dynamics. Fourth, taking the participation of multiple parties as an opportunity to study the endogenous mechanisms for the development of the NFPR system. The key to solving the sustainable management dilemma of natural forests is to stimulate the spontaneous participation behavior of multiple parties and to form an endogenous driving force mechanism. Future research can further reveal this mechanism by applying institutional rules and constructing reasonable evaluation criteria, thus trying to achieve sustainable development of natural forest resources.

## 7. Conclusions

This research takes the participation of multiple parties in NFPR as the research route, adopts the IAD framework, and systematically analyzes the relevant literature in terms of the factors affecting the participation of multiple parties in NFPR and the effects of the outcomes, and constructs the mechanism of multiple parties' participation in NFPR. It was found that the external variables affecting the participation of multiple parties in NFPR mainly include four categories: characteristics of multiple participants, biophysical conditions, attributes of the community, and rules-in-use, and each category contains different specific influencing factors, which form an organic whole through interconnection, action, and regulation with each other. The participation behavior of multiple parties results from the joint action of various external variables, and it is difficult for a single variable or factor to achieve the sustainable development of multiple participants in NFPR. Rules-in-use plays an important role in regulating the participation of multiple parties in the process of NFPR. Changes in rules-in-use can be used to adapt multiple participants' characteristics, biophysical conditions, and attributes of the community, and promote their participation in NFPR through the interaction modes such as forest certifications, forest carbon sink trading, and natural commercial forest redemption, thus improving the quality of natural forests. This research not only provides a theoretical basis for NFPR in China, but also provides an empirical reference for NFPR in other countries and regions.

Future researchers may focus on the development of the IAD framework, cutting-edge cross-disciplinary research, and the study of the dynamics of spatial and temporal differences, as well as the intrinsic motivation of multiple participating parties brought by rules-in-use to promote NFPR and achieve sustainable management.

**Author Contributions:** Conceptualization, J.K. and C.Z.; methodology, J.K. and R.S.; formal analysis, J.K. and R.S.; resources, J.K. and C.Z.; writing—original draft preparation, J.K. and R.S.; writing—review and editing, C.Z. and Y.M.; supervision, C.Z.; project administration, C.Z.; funding acquisition, J.K. and C.Z. All authors have read and agreed to the published version of the manuscript.

**Funding:** This research was funded by the Project of Economic Development Research Center of National Forestry and Grassland Administration, "Research on investment and financing Policies for Natural Forest Restoration" (grant number JYCL-2020-00021), the National Social Science Foundation of China (grant number 21BJY247).

**Data Availability Statement:** The data presented in this study are available from the corresponding author upon request. The data are not publicly available due to the national law on the restriction of privacy.

**Conflicts of Interest:** The authors declare no conflict of interest.

## Appendix A

**Table A1.** Summary of influencing factors and outcomes of multiple parties' participation in NFPR.

| Type | External Variables | | Effects | Spatial Scale | Time Scale | Reference |
|---|---|---|---|---|---|---|
| Multiple Participants' Characteristics | Characteristics of Farmers | Farmer population | + | Yichun City, Heilongjiang Province, China | 2003–2018 | Zhao, X. (2021) |
| | | | − | Tropical forests | 1998 | Arild, A et al. (1999) |
| | | | − | Tropical forests | 1986–1999 | Geist, H. et al. (2001) |
| | | | − | Northern Ecuadorian Amazon | 1990–1999 | Alisson, F. B. et al. (2005) |
| | | | + | Heilongjiang Province, China | 1977–2013 | Shi, M. et al. (2017) |
| | | | + | China | 1995–2005 | Chen, X. (2007) |
| | | | + | China | 1973–2009 | Shi, C. (2010) |
| | | Farmers' income | + | Wulong County, Chongqing Municipality | 2009 | Zhi, L. et al. (2013) |
| | | Farmers' willingness | + | Heilongjiang Province, China | 1996–2013 | Zheng, L. (2015) |
| | | Part-time farmers | − | China | 1997–2006 | Feng, Q. et al. (2010) |
| | Characteristics of Forest Management | | unclear | Adaba and Dodola districts of West Arsi zone | 2004–2012 | Goytom, A. et al. (2019) |
| | | | unclear | Central Senegal | 2006–2010 | Robinson, E. et al. (2021) |
| | Characteristics of Government Apparatus | | + | Northern Pakistan | 2019 | Ali, S. et al. (2021) |
| | | | + | Xiushan County, Chongqing Municipality | 2007–2011 | Jin, H. (2012) |
| | | | + | Qinling Nature Reserve, China | 2006–2012 | Liu, J. et al. (2013) |
| | Characteristics of Other Participants | | + | Heilongjiang Province, China | 1998–2013 | Zhu, Y. (2015) |
| Biophysical Conditions | Forest Land Conditions | | + | Heilongjiang Province, China | 1996–2013 | Zheng, L. (2015) |
| | | | + | Heilongjiang Province, China | 1977–2013 | Shi, M. et al. (2017) |
| | | | + | Beijing-Tianjin-Hebei region, China | 2000–2018 | Li, L. et al. (2021) |

**Table A1.** *Cont.*

| Type | External Variables | Effects | Spatial Scale | Time Scale | Reference |
|---|---|---|---|---|---|
| Biophysical Conditions | Forest Conditions | + | Heilongjiang Province, China | 1996–2013 | Zheng, L. (2015) |
| | | + | South China | 1977–2013 | Yan, R. et al. (2020) |
| | Climate Conditions | + | Da Hinggan Mountains, China | 1990–2015 | Zheng, S. et al. (2021) |
| | | + | Heilongjiang Province, China | 1996–2013 | Zheng, L. (2015) |
| | Geographic Location | unclear | China | 1995–2005 | Chen, X. (2007) |
| | | unclear | Western China | 2002–2012 | Ke, S. et al. (2015) |
| | | unclear | Xiaolongshan region, Gansu Province, China | 2000–2010 | Sun, C. et al. (2017) |
| | Natural Disasters in Forest | – | Heilongjiang Province, China | 1996–2013 | Zheng, L. (2015) |
| | | unclear | Yichun City, Heilongjiang Province, China | 2003–2018 | Zhao, X. (2021) |
| Attributes of the Community | Economic Development Level | – | China | 1973–2009 | Shi, C. (2010) |
| | | + | Heilongjiang Province, China | 2000–2015 | Zhou, M. (2017) |
| | Industrial Restructuring | + | Northeast China | 1993–2014 | Chen, Y. et al. (2017) |
| | | – | Heilongjiang Province, China | 1976–2015 | Zhang, B. (2020) |
| | Forest Carbon Sink Market | + | China | 2004–2013 | Jiang, X. (2016) |
| | | + | China | 2010–2016 | Zhao, H. (2019) |
| | | + | Heilongjiang Province, China | 2005–2014 | Chen, L. et al. (2021) |
| | Agricultural Production Efficiency | + | Sri Lanka | 2004 | Prabodh, I. (2005) |
| | | + | National scale | 2009–2012 | Ricardo, G. et al. (2013) |
| | | + | Nepal | 2013 | Narendra, C. et al. (2014) |
| | Customs and Culture | + | China | 1995–2005 | Chen, X. (2007) |
| Rules-in-use | Direct Rules — Direct investment | + | Heilongjiang Province, China | 2000–2015 | Zhou, M. (2017) |
| | | + | China | 1998-2003 | Lv, J. (2005) |
| | Direct Rules — Logging ban policy | + | China | 2004–2013 | Jiang, X. (2016) |
| | | + | China | 2000–2014 | Zhu, Z. et al. (2018) |
| | | + | Northeast China | 2016 | Zou, Y. et al. (2020) |
| | | – | Yichun City, Heilongjiang Province, China | 2003–2018 | Zhao, X. (2021) |
| | | – | China | 2017 | Ke, S. et al. (2018) |

**Table A1.** *Cont.*

| Type | External Variables | | Effects | Spatial Scale | Time Scale | Reference |
|---|---|---|---|---|---|---|
| Rules-in-use | Indirect Rules | | + | Yichun City, Heilongjiang Province, China | 2003–2018 | Zhao, X. (2021) |
| | | | + | China | 2019 | Lin, X. et al. (2020) |
| | Organizational Rules | Ecosystem management | + | Moshao forest farm of Huitong county, China | 1998–2013 | Dai, E. et al. (2020) |
| | | | + | Heilongjiang Province, China | 1998–2013 | Zhu, Y. (2015) |
| | | Contract management | + | Beijing, China | 2009–2014 | Li, C. et al. (2021) |
| | | Division of management zones | unclear | Northeast China | 2000–2010 | Wei, Y. et al. (2014) |
| | | Collective forest tenure reform | + | China | 1989–2018 | Liu, S. et al. (2021) |
| | | | + | China | 1990–2009 | Zhang, H. (2012) |
| | Incentive Rules | | + | Africa, Asia and Latin America | 1991 | Fortmann, L. et al. (1991) |
| | | | + | China | 1949–2013 | Hu, A. et al. (2014) |
| | | | + | Wuling Mountain Areas, China | 2012 | Li, X. et al. (2015) |
| | | | + | Ningxia Province, China | 2015–2019 | Wang, Y. (2020) |
| Outcomes | Direct Outcomes | Ecological benefit | | Yichun City, Heilongjiang Province, China | 2019 | Qin, L. et al. (2021) |
| | | | | Shanxi Province, China | 2016 | Fan, L. (2019) |
| | | | | Northeast China and Inner Mongolia | 2000–2015 | Huang, L. et al. (2017) |
| | | Ecosystem services restoration improvement | | Changbai Mountain, China | 2000–2015 | Wang, H. et al. (2017) |
| | | | | Da Hinggan Mountains, China | 1990–2015 | Zheng, S. et al. (2021) |
| | | | | Xinjiang province, China | 2000–2015 | Lan, J. et al. (2018) |
| | | Carbon stock and carbon sink increase | | Upper Reaches of Yangtze river | 1988–2008 | Guo, Y. et al. (2015) |
| | | | | Heilongjiang Province, China | 1973–2013 | Zhang, C. et al. (2018) |
| | | | | China | 1999–2018 | Zhang, Y. et al. (2021) |
| | | | | Beijing, China | 2009–2014 | Li, C. et al. (2021) |

**Table A1.** *Cont.*

| Type | External Variables | Effects | Spatial Scale | Time Scale | Reference |
|---|---|---|---|---|---|
| Outcomes | Indirect Outcomes | | Northeast China | 2012–2017 | Geng, Y. et al. (2021) |
| | Farmers' income | | Western China | 2003–2011 | Zang, L. (2014) |
| | Farmer employment | | Yichun City, Heilongjiang Province, China | 2008, 2013 | Song, X. et al. (2018) |
| | Forest industry enterprise development | | Northeast China | 2000–2015 | Deng, Y. et al. (2019) |
| | Environmental health | | China | 1996–2015 | Farooq, M.U. et al. (2019) |
| | | | Cambodia | 2000–2014 | Pienkowski, T. et al. (2017) |

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
