# Peer review of "What Promotes Natural Forest Protection and Restoration? Insights from the Perspective of Multiple Parties"

_forests, doi:10.3390/f14020339_

Round 1

Reviewer 1 Report

Overall comments: The manuscript had well organized and writing skills also very good. Paper need minor revision. Some of the specific comments are discussed bellow

L 39 Before 1998, the ecological functions of natural forests were severely degraded due to over-harvesting and unreasonable management...Why its so can you explain, any citation for this statement.

L 46 NFPR, since it’s already abbreviated in abstract so no need to expand the same, follow the same in throughout manuscript.

L161 [25–27]With the NFPR... replace as [25-27] with...please keep a space with [xx], Please do follow MDPI citation pattern throughout the manuscript.

L 187 which will lead to changes in family income structure and affect the participation behavior of NFPR

Will can replace as “can be” and behaviour as well.

L316 Guo’s research discussed...where is the citation?

L 494 /521 Figure 2/3 Can you pls enhance it?

Conclusion section is too big, can you make it compressive. Spell checks are highly appreciated.

Reviewer 2 Report

Dear Authors,

The manuscript presents a distinct issue for China regarding the "sustainable management dilemma" that challenges the relationships of organizations and their economic, natural, and social environments for the conservation of natural forests and for restructuring to demonstrate an efficient and sustainable market. The text shows the NFPR from the viewpoint of multiple subjects, which can offer corresponding theoretical support for developing and improving this system (NFPR) and provide characteristic Chinese experiences for other nations. The text brings the complex mechanisms of various influencing factors of the NFPR.

The authors use the IAD Framework as its unit of analysis of the political network or arena of action to clarify the mechanism driving the participation of multiple subjects and explain the rationale behind their involvement in the NFPR.

When reading, the text is informative. However, the Methodology, is too long and needs to be restructured. I suggested putting the textual part as a complement.

I did not find the topics that guided the Results and show the Discussions. It needs to be verified by the authors.

The conclusion brings elements of Discussion. The topic Conclusion is too long and needs to be rewritten.

Abstract

The abstract needs to be more explicit. The beginning puts "imperfect system of Natural Forest Protection and Restoration," but why?

What does "sustainable management dilemma" mean? What are "multiple subjects' characteristics"? This argument needs to be better written. The reader starts reading the abstract and does not understand how the manuscript will be final (objective, hypothesis, clear results, and conclusion). Rewrite, please.

Introduction

Line 31: "The forest stock is" the forest stock is wood, carbon, biomass. What does this information mean for work? Please clarify.

Lines 35 and 36: "Natural forests, originated in a natural state, have not undergone human disturbance." This phrase is controversial because no matter how well-preserved the forest is, human beings have been disturbed or may have been disturbed once. Review, please.

Line 43: A management of participation can complement the dilemma suggested and thus add to Sustainable Management? Would this be an essential question for the manuscript?

Lines 58-65: This part shows essential information that may be in the abstract to clarify to the reader a part of the problem proposed in the manuscript. That's the dilemma! Review, please.

Line 109: The Elinor Ostrom date is missing.

Methodology

Lines 147 to 450: The information needs to be summarized or created as a diagram and pass the text to annex/complementary material for the reader's understanding.

There should be a Results topic and then Discussion.

Conclusions

The conclusions of the manuscript are too long. Many parts are part of a discussion I did not find the topic in the text. Restructure, please.
